# Q-LDA: Uncovering Latent Patterns in Text-based Sequential Decision Processes

**Jianshu Chen**[*]**, Chong Wang**[†]**, Lin Xiao**[*]**, Ji He**[‡]**, Lihong Li**[†] **and Li Deng**[‡]

[*]Microsoft Research, Redmond, WA, USA
`{jianshuc,lin.xiao}@microsoft.com`
[†]Google Inc., Kirkland, WA, USA[*]
`{chongw,lihong}@google.com`
[‡]Citadel LLC, Seattle/Chicago, USA
`{Ji.He,Li.Deng}@citadel.com`

## Abstract

In sequential decision making, it is often important and useful for end users to understand the underlying patterns or causes that lead to the corresponding decisions. However, typical deep reinforcement learning algorithms seldom provide such information due to their black-box nature. In this paper, we present a probabilistic model, Q-LDA, to uncover latent patterns in text-based sequential decision processes. The model can be understood as a variant of latent topic models that are tailored to maximize total rewards; we further draw an interesting connection between an approximate maximum-likelihood estimation of Q-LDA and the celebrated Q-learning algorithm. We demonstrate in the text-game domain that our proposed method not only provides a viable mechanism to uncover latent patterns in decision processes, but also obtains state-of-the-art rewards in these games.

## 1  Introduction

Reinforcement learning [21] plays an important role in solving sequential decision making problems, and has seen considerable successes in many applications [16, 18, 20]. With these methods, however, it is often difficult to understand or examine the underlying patterns or causes that lead to the sequence of decisions. Being more interpretable to end users can provide more insights to the problem itself and be potentially useful for downstream applications based on these results [5].

To investigate new approaches to uncovering underlying patterns of a text-based sequential decision process, we use text games (also known as interactive fictions) [11, 19] as the experimental domain. Specifically, we focus on choice-based and hypertext-based games studied in the literature [11], where both the action space and the state space are characterized in natural languages. At each time step, the decision maker (i.e., *agent*) observes one text document (i.e., *observation text*) that describes the current observation of the game environment, and several text documents (i.e., *action texts*) that characterize different possible actions that can be taken. Based on the history of these observations, the agent selects one of the provided actions and the game transits to a new state with an *immediate reward*. This game continues until the agent reaches a final state and receives a *terminal reward*.

In this paper, we present a probabilistic model called *Q-LDA* that is tailored to maximize total rewards in a decision process. Specially, observation texts and action texts are characterized by two separate topic models, which are variants of latent Dirichlet allocation (LDA) [4]. In each topic model, topic proportions are chained over time to model the dependencies for actions or states. And

---

[*]The work was done while Chong Wang, Ji He, Lihong Li and Li Deng were at Microsoft Research.

these proportions are partially responsible for generating the immediate/terminal rewards. We also show an interesting connection between the maximum-likelihood parameter estimation of the model and the Q-learning algorithm [22, 18]. We empirically demonstrate that our proposed method not only provides a viable mechanism to uncover latent patterns in decision processes, but also obtains state-of-the-art performance in these text games.

**Contribution.** The main contribution of this paper is to seamlessly integrate topic modeling with Q-learning to uncover the latent patterns and interpretable causes in text-based sequential decision-making processes. Contemporary deep reinforcement learning models and algorithms can seldom provide such information due to their black-box nature. To the best of our knowledge, there is no prior work that can achieve this and learn the topic model in an end-to-end fashion to maximize the long-term reward.

**Related work.** Q-LDA uses variants of LDA to capture observation and action texts in text-based decision processes. In this model, the dependence of immediate reward on the topic proportions is similar to supervised topic models [3], and the chaining of topic proportions over time to model long-term dependencies on previous actions and observations is similar to dynamic topic models [6]. The novelty in our approach is that the model is estimated in a way that aims to maximize long-term reward, thus producing near-optimal policies; hence it can also be viewed as a topic-model-based reinforcement-learning algorithm. Furthermore, we show an interesting connection to the DQN variant of Q-learning [18]. The text-game setup used in our experiment is most similar to previous work [11] in that both observations and actions are described by natural languages, leading to challenges in both representation and learning. The main difference from that previous work is that those authors treat observation-texts as Markovian states. In contrast, our model is more general, capturing both partial observability and long-term dependence on observations that are common in many text-based decision processes such as dialogues. Finally, the choice of reward function in Q-LDA share similarity with that in Gaussian process temporal difference methods [9].

**Organization.** Section 2 describes the details of our probabilistic model, and draws a connection to the Q-learning algorithm. Section 3 presents an end-to-end learning algorithm that is based on mirror descent back-propagation. Section 4 demonstrates the empirical performance of our model, and we conclude with discussions and future work in Section 5.

## 2 A Probabilistic Model for Text-based Sequential Decision Processes

In this section, we first describe text games as an example of sequential decision processes. Then, we describe our probabilistic model, and relate it to a variant of Q-learning.

### 2.1 Sequential decision making in text games

Text games are an episodic task that proceeds in discrete time steps $t \in \{1, \ldots, T\}$, where the length $T$ may vary across different *episodes*. At time step $t$, the agent receives a text document of $N$ words describing the current observation of the environment: $w_t^S \triangleq \{w_{t,n}^S\}_{n=1}^N$.[2] We call these words *observation text*. The agent also receives $A_t$ text documents, each of which describes a possible action that the agent can take. We denote them by $w_t^a \triangleq \{w_{t,n}^a\}_{n=1}^N$ with $a \in \{1, \ldots, A_t\}$, where $A_t$ is the number of feasible actions and it could vary over time. We call these texts *action texts*. After the agent takes one of the provided actions, the environment transits to time $t + 1$ with a new state and an immediate reward $r_t$; both dynamics and reward generation may be stochastic and unknown. The new state then reveals a new observation text $w_{t+1}^S$ and several action texts $w_{t+1}^a$ for $a \in \{1, \ldots, A_{t+1}\}$. The transition continues until the end of the game at step $T$ when the agent receives a *terminal reward* $r_T$. The reward $r_T$ depends on the ending of the story in the text game: a good ending leads to a large positive reward, while bad endings negative rewards.

The goal of the agent is to maximize its cumulative reward by acting optimally in the environment. At step $t$, given all observation texts $w_{1:t}^S$, all action texts $w_{1:t}^A \triangleq \{w_{1:t}^a : \forall a\}$, previous actions $a_{1:t-1}$ and rewards $r_{1:t-1}$, the agent is to find a *policy*, $\pi(a_t | w_{1:t}^S, w_{1:t}^A, a_{1:t-1}, r_{1:t-1})$, a conditional

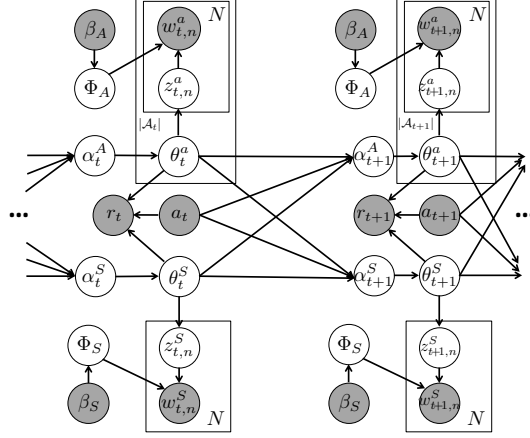

Figure 1: Graphical model representation for the studied sequential decision process. The bottom section shows the observation topic models, which share the same topics in $\Phi_S$, but the topic distributions $\theta_t^S$ changes with time $t$. The top section shows the action topic models, sharing the same action topics in $\Phi_A$, but with time varying topic distribution $\theta_t^a$ for each $a \in A_t$. The middle section shows the dependence of variables between consecutive time steps. There are no plates for the observation text (bottom part of the figure) because there is only one observation text document at each time step. We follow the standard notation for graphical models by using shaded circles as observables. Since the topic distributions $\theta_t^S$ and $\theta_t^a$ and the Dirichlet parameters $\alpha_t^S$ and $\alpha_t^A$ (except $\alpha_1^S$ and $\alpha_1^A$) are not observable, we need to use their MAP estimate to make end-to-end learning feasible; see Section 3 for details. The figure characterizes the general case where rewards appear at each time step, while in our experiments the (non-zero) rewards only appear at the end of the games.

probability of selecting action $a_t$, that maximizes the expected long-term reward $\mathbb{E}\{\sum_{\tau=t}^{T} \gamma^{\tau-t} r_\tau\}$, where $\gamma \in (0,1)$ is a discount factor. In this paper, for simplicity of exposition, we focus on problems where the reward is nonzero only in the final step $T$. While our algorithm can be generalized to the general case (with greater complexity), this special case is an important case of RL (e.g., [20]). As a result, the policy is independent of $r_{1:t-1}$ and its form is simplified to $\pi(a_t | w_{1:t}^S, w_{1:t}^A, a_{1:t-1})$.

The problem setup is similar to previous work [11] in that both observations and actions are described by natural languages. For actions described by natural languages, the action space is inherently discrete and large due to the exponential complexity with respect to sentence length. This is different from most reinforcement learning problems where the action spaces are either small or continuous. Here, we take a probabilistic modeling approach to this challenge: the observed variables—observation texts, action texts, selected actions, and rewards—are assumed to be generated from a probabilistic latent variable model. By examining these latent variables, we aim to uncover the underlying patterns that lead to the sequence of the decisions. We then show how the model is related to Q-learning, so that estimation of the model leads to reward maximization.

## 2.2 The Q-LDA model

The graphical representation of our model, Q-LDA, is depicted in Figure 1. It has two instances of topic models, one for observation texts and the other for action texts. The basic idea is to chain the topic proportions ($\theta$s in the figure) in a way such that they can influence the topic proportions in the future, thus capturing long-term effects of actions. Details of the generative models are as follows.

For the observation topic model, we use the columns of $\Phi_S \sim \mathrm{Dir}(\beta_S)^3$ to denote the topics for the observation texts. For the action topic model, we use the columns of $\Phi_A \sim \mathrm{Dir}(\beta_A)$ to denote the topics for the action texts. We assume these topics do not change over time. Given the initial topic proportion Dirichlet parameters—$\alpha_1^S$ and $\alpha_1^A$ for observation and action texts respectively—the Q-LDA proceeds sequentially from $t = 1$ to $T$ as follows (see Figure 1 for all latent variables).

---

³$\Phi_S$ is a word-by-topic matrix. Each column is drawn from a Dirichlet distribution with hyper-parameter $\beta_S$, representing the word-emission probabilities of the corresponding topic. $\Phi_A$ is similarly defined.

1. Draw observation text $w_t^S$ as follows,
   (a) Draw observation topic proportions $\theta_t^S \sim \text{Dir}(\alpha_t^S)$.
   (b) Draw all words for the observation text $w_t^S \sim \text{LDA}(w_t^S|\theta_t^S, \Phi_S)$, where $\text{LDA}(\cdot)$ denotes the standard LDA generative process given its topic proportion $\theta_t^S$ and topics $\Phi_S$ [4]. The latent variable $z_{t,n}^S$ indicates the topic for the word $w_{t,n}^S$.
2. For $a = 1, ..., A_t$, draw action text $w_t^a$ as follows,
   (a) Draw action topic proportions $\theta_t^a \sim \text{Dir}(\alpha_t^A)$.
   (b) Draw all words for the $a$-th action text using $w_t^a \sim \text{LDA}(w_t^a|\theta_t^a, \Phi_A)$, where the latent variable $z_{t,n}^a$ indicates the topic for the word $w_{t,n}^a$.
3. Draw the action: $a_t \sim \pi_b(a_t|w_{1:t}^S, w_{1:t}^A, a_{1:t-1})$, where $\pi_b$ is an *exploration policy* for data collection. It could be chosen in different ways, as discussed in the experiment Section 4. After model learning is finished, a greedy policy may be used instead (c.f., Section 3).
4. The immediate reward $r_t$ is generated according to a Gaussian distribution with mean function $\mu_r(\theta_t^S, \theta_t^{a_t}, U)$ and variance $\sigma_r^2$:

$$r_t \sim \mathcal{N}\left(\mu_r(\theta_t^S, \theta_t^{a_t}, U), \sigma_r^2\right) . \tag{1}$$

   Here, we defer the definitions of $\mu_r(\theta_t^S, \theta_t^{a_t}, U)$ and its parameter $U$ to the next section, where we draw a connection between likelihood-based learning and Q-learning.
5. Compute the topic proportions Dirichlet parameters for the next time step $t+1$ as

$$\alpha_{t+1}^S = \sigma\left(W_{SS}\theta_t^S + W_{SA}\theta_t^{a_t} + \alpha_1^S\right), \quad \alpha_{t+1}^A = \sigma\left(W_{AS}\theta_t^S + W_{AA}\theta_t^{a_t} + \alpha_1^A\right), \tag{2}$$

   where $\sigma(x) \triangleq \max\{x, \epsilon\}$ with $\epsilon$ being a small positive number (e.g., $10^{-6}$), $a_t$ is the action selected by the agent at time $t$, and $\{W_{SS}, W_{SA}, W_{AS}, W_{AA}\}$ are the model parameters to be learned. Note that, besides $\theta_t^S$, the only topic proportions from $\{\theta_t^a\}_{a=1}^{A_t}$ that will influence $\alpha_{t+1}^S$ and $\alpha_{t+1}^A$ is $\theta_t^{a_t}$, i.e., the one corresponding to the chosen action $a_t$. Furthermore, since $\theta_t^S$ and $\theta_t^{a_t}$ are generated according to $\text{Dir}(\alpha_t^S)$ and $\text{Dir}(\alpha_t^A)$, respectively, $\alpha_{t+1}^S$ and $\alpha_{t+1}^A$ are (implicitly) chained over time via $\theta_t^S$ and $\theta_t^{a_t}$ (c.f. Figure 1).

This generative process defines a joint distribution $p(\cdot)$ among all random variables depicted in Figure 1. Running this generative process—step 1 to 5 above for $T$ steps until the game ends—produces one episode of the game. Now suppose we already have $M$ episodes. In this paper, we choose to directly learn the conditional distribution of the rewards given other observations. By learning the model in a *discriminative* manner [2, 7, 12, 15, 23], we hope to make better predictions of the rewards for different actions, from which the agent could obtain the best policy for taking actions. This can be obtained by applying Bayes rule to the joint distribution defined by the generative process. Let $\Theta$ denote all model parameters: $\Theta = \{\Phi_S, \Phi_A, U, W_{SS}, W_{SA}, W_{AS}, W_{AA}\}$. We have the following loss function

$$\min_{\Theta} \left\{ -\ln p(\Theta) - \sum_{i=1}^{M} \ln p\left(r_{1:T_i}|w_{1:T_i}^S, w_{1:T_i}^A, a_{1:T_i}, \Theta\right) \right\}, \tag{3}$$

where $p(\Theta)$ denotes a prior distribution of the model parameters (e.g., Dirichlet parameters over $\Phi_S$ and $\Phi_A$), and $T_i$ denotes the length of the $i$-th episode. Let $K_S$ and $K_A$ denote the number of topics for the observation texts and action texts, and let $V_S$ and $V_A$ denote the vocabulary sizes for the observation texts and action texts, respectively. Then, the total number of learnable parameters for Q-LDA is: $V_S \times K_S + V_A \times K_A + K_A \times K_S + (K_S + K_A)^2$.

We note that a good model learned through Eq. (3) may predict the values of rewards well, but might not imply the best policy for the game. Next, we show by defining the appropriate mean function for the rewards, $\mu_r(\theta_t^S, \theta_t^{a_t}, U)$, we can achieve both. This closely resembles Q-learning [21, 22], allowing us to effectively learn the policy in an iterative fashion.

## 2.3 From Q-LDA to Q-learning

Before relating Q-LDA to Q-learning, we first give a brief introduction to the latter. Q-learning [22, 18] is a reinforcement learning algorithm for finding an optimal policy in a Markov decision process (MDP) described by $(\mathcal{S}, \mathcal{A}, \mathcal{P}, r, \gamma)$, where $\mathcal{S}$ is a state space, $\mathcal{A}$ is an action space, and $\gamma \in (0, 1)$ is a discount factor. Furthermore, $\mathcal{P}$ defines a transition probability $p(s'|s, a)$ for going to the next

state $s' \in \mathcal{S}$ from the current state $s \in \mathcal{S}$ after taking action $a \in \mathcal{A}$, and $r(s,a)$ is the immediate reward corresponding to this transition. A policy $\pi(a|s)$ in an MDP is defined to be the probability of taking action $a$ at state $s$. Let $s_t$ and $a_t$ be the state and action at time $t$, and let $r_t = r(s_t, a_t)$ be the immediate reward at time $t$. An optimal policy is the one that maximizes the expected long-term reward $\mathbb{E}\{\sum_{t=1}^{+\infty} \gamma^{t-1} r_t\}$. Q-learning seeks to find the optimal policy by estimating the Q-function, $Q(s,a)$, defined as the expected long-term discounted reward for taking action $a$ at state $s$ and then following an optimal policy thereafter. It satisfies the Bellman equation [21]

$$Q(s,a) = \mathbb{E}\{r(s,a) + \gamma \cdot \max_b Q(s',b)|s,a\}, \tag{4}$$

and directly gives the optimal action for any state $s$: $\arg\max_a Q(s,a)$.

Q-learning solves for $Q(s,a)$ iteratively based on observed state transitions. The basic Q-learning [22] requires storing and updating the values of $Q(s,a)$ for all state–action pairs in $\mathcal{S} \times \mathcal{A}$, which is not practical when $\mathcal{S}$ and $\mathcal{A}$ are large. This is especially true in our text games, where they can be exponentially large. Hence, $Q(s,a)$ is usually approximated by a parametric function $Q_\theta(s,a)$ (e.g., neural networks [18]), in which case the model parameter $\theta$ is updated by:

$$\theta \leftarrow \theta + \eta \cdot \nabla_\theta Q_\theta \cdot (d_t - Q_\theta(s_t, a_t)), \tag{5}$$

where $d_t \triangleq r_t + \gamma \cdot \max_{a'} Q_{\theta'}(s_{t+1}, a')$ if $s_t$ nonterminal and $d_t \triangleq r_t$ otherwise, and $\theta'$ denotes a delayed version of the model parameter updated periodically [18]. The update rule (5) may be understood as applying stochastic gradient descent (SGD) to a regression loss function $J(\theta) \triangleq \mathbb{E}[d_t - Q_\theta(s,a)]^2$. Thus, $d_t$ is the *target*, computed from $r_t$ and $Q_{\theta'}$, for the prediction $Q_\theta(s_t, a_t)$.

We are now ready to define the mean reward function $\mu_r$ in Q-LDA. First, we model the Q-function by $Q(\theta_t^S, \theta_t^a) = (\theta_t^a)^T U \theta_t^S$, where $U$ is the same parameter as the one in (1).[4] This is different from typical deep RL approaches, where black-box models like neural networks are used. In order to connect our probabilistic model to Q-learning, we define the mean reward function as follows,

$$\mu_r(\theta_t^S, \theta_t^{a_t}, U) = Q(\theta_t^S, \theta_t^{a_t}) - \gamma \cdot \mathbb{E}\left[\max_b Q(\theta_{t+1}^S, \theta_{t+1}^b)|\theta_t^S, \theta_t^{a_t}\right] \tag{6}$$

Note that $\mu_r$ remains as a function of $\theta_t^S$ and $\theta_t^{a_t}$ since the second term in the above expression is a conditional expectation given $\theta_t^S$ and $\theta_t^{a_t}$. The definition of the mean reward function in Eq. (6) has a strong relationship with the Bellman equation (4) in Q-learning; it relates the long-term reward $Q(\theta_t^S, \theta_t^{a_t})$ to the mean immediate reward $\mu_r$ in the same manner as the Bellman equation (4). To see this, we move the second term on the right-hand side of (6) to the left, and make the identification that $\mu_r$ corresponds to $\mathbb{E}\{r(s,a)\}$ since both of them represent the mean immediate reward. The resulting equation share a same form as the Bellman equation (4). With the mean function $\mu_r$ defined above, we show in Appendix B that the loss function (3) can be approximated by the one below using the maximum a posteriori (MAP) estimate of $\theta_t^S$ and $\theta_t^{a_t}$ (denoted as $\hat{\theta}_t^S$ and $\hat{\theta}_t^{a_t}$, respectively):

$$\min_\Theta \left\{ -\ln p(\Phi_S|\beta_S) - \ln p(\Phi_A|\beta_A) + \sum_{i=1}^M \sum_{t=1}^{T_i} \frac{1}{2\sigma_r^2}\left[d_t - Q(\hat{\theta}_t^S, \hat{\theta}_t^{a_t})\right]^2 \right\} \tag{7}$$

where $d_t = r_t + \gamma \max_b Q(\hat{\theta}_{t+1}^S, \hat{\theta}_{t+1}^b)$ for $t < T_i$ and $d_t = r_t$ for $t = T_i$. Observe that the first two terms in (7) are regularization terms coming from the Dirichlet prior over $\Phi_S$ and $\Phi_A$, and the third term shares a similar form as the cost $J(\theta)$ in Q-learning; it can also be interpreted as a regression problem for estimating the Q-function, where the target $d_t$ is constructed in a similar manner as Q-learning. Therefore, optimizing the discriminative objective (3) leads to a variant of Q-learning. After learning is finished, we can obtain the greedy policy by taking the action that maximizes the Q-function estimate in any given state.

We also note that we have used the MAP estimates of $\theta_t^S$ and $\theta_t^{a_t}$ due to the intractable marginalization of the latent variables [14]. Other more advanced approximation techniques, such as Markov Chain Monte Carlo (MCMC) [1] and variational inference [13] can also be used, and we leave these explorations as future work.

## 3  End-to-end Learning by Mirror Descent Back Propagation

**Algorithm 1** The training algorithm by mirror descent back propagation

---

1: **Input:** $D$ (number of experience replays), $J$ (number of SGD updates), and learning rate.
2: Randomly initialize the model parameters.
3: **for** $m = 1, \ldots, D$ **do**
4:     Interact with the environment using a behavior policy $\pi_b^m(a_t|x_{1:t}^S, x_{1:t}^A, a_{1:t-1})$ to collect $M$ episodes of data $\{w_{1:T_i}^S, w_{1:T_i}^A, a_{1:T_i}, r_{1:T_i}\}_{i=1}^M$ and add them to $\mathcal{D}$.
5:     **for** $j = 1, \ldots, J$ **do**
6:         Randomly sample an episode from $\mathcal{D}$.
7:         For the sampled episode, compute $\hat{\theta}_t^S$, $\hat{\theta}_t^a$ and $Q(\hat{\theta}_t^S, \hat{\theta}_t^a)$ with $a = 1, \ldots, A_t$ and $t = 1, \ldots, T_i$ according to Algorithm 2.
8:         For the sampled episode, compute the stochastic gradients of (7) with respect to $\Theta$ using back propagation through the computational graph defined in Algorithm 2.
9:         Update $\{U, W_{SS}, W_{SA}, W_{AS}, W_{AA}\}$ by stochastic gradient descent and update $\{\Phi_S, \Phi_A\}$ using stochastic mirror descent.
10:     **end for**
11: **end for**

---

---

**Algorithm 2** The recursive MAP inference for one episode

---

1: **Input:** $\alpha_1^S$, $\alpha_1^A$, $L$, $\delta$, $x_t^S$, $\{x_t^a : a = 1, \ldots, A_t\}$ and $a_t$, for all $t = 1, \ldots, T_i$.
2: Initialization: $\hat{\alpha}_1^S = \alpha_1^S$ and $\hat{\alpha}_1^A = \alpha_1^A$
3: **for** $t = 1, \ldots, T_i$ **do**
4:     Compute $\hat{\theta}_t^S$ by repeating $\hat{\theta}_t^S \leftarrow \frac{1}{C}\hat{\theta}_t^S \odot \exp\left(\delta\left[\Phi_S^T \frac{x_t^S}{\Phi_S\hat{\theta}_t^S} + \frac{\hat{\alpha}_t^S - \mathbb{1}}{\hat{\theta}_t^S}\right]\right)$ for $L$ times with initialization $\hat{\theta}_t^S \propto \mathbb{1}$, where $C$ is a normalization factor.
5:     Compute $\hat{\theta}_t^a$ for each $a = 1, \ldots, A_t$ by repeating $\hat{\theta}_t^a \leftarrow \frac{1}{C}\hat{\theta}_t^a \odot \exp\left(\delta\left[\Phi_A^T \frac{x_t^a}{\Phi_A\hat{\theta}_t^a} + \frac{\hat{\alpha}_t^A - \mathbb{1}}{\hat{\theta}_t^a}\right]\right)$ for $L$ times with initialization $\hat{\theta}_t^a \propto \mathbb{1}$, where $C$ is a normalization factor.
6:     Compute $\hat{\alpha}_{t+1}^S$ and $\hat{\alpha}_{t+1}^A$ from $\hat{\theta}_t^S$ and $\hat{\theta}_t^{a_t}$ according to (11).
7:     Compute the Q-values: $Q(\hat{\theta}_t^S, \hat{\theta}_t^a) = (\hat{\theta}_t^a)^T U \hat{\theta}_t^S$ for $a = 1, \ldots, A_t$.
8: **end for**

---

In this section, we develop an end-to-end learning algorithm for Q-LDA, by minimizing the loss function given in (7). As shown in the previous section, solving (7) leads to a variant of Q-learning, thus our algorithm could be viewed as a reinforcement-learning algorithm for the proposed model.

We consider learning our model with experience replay [17], a widely used technique in recent state-of-the-art systems [18]. Specifically, the learning process consists of multiple stages, and at each stage, the agent interacts with the environment using a fixed exploration policy $\pi_b(a_t|x_{1:t}^S, x_{1:t}^A, a_{1:t-1})$ to collect $M$ episodes of data $\{w_{1:T_i}^S, w_{1:T_i}^A, a_{1:T_i}, r_{1:T_i}\}_{i=1}^M$ and saves them into a *replay memory* $\mathcal{D}$. (We will discuss the choice of $\pi_b$ in section 4.) Under the assumption of the generative model Q-LDA, our objective is to update our estimates of the model parameters in $\Theta$ using $\mathcal{D}$; the updating process may take several randomized passes over the data in $\mathcal{D}$. A stage of such learning process is called one *replay*. Once a replay is done, we let the agent use a new behavior policy $\pi_b'$ to collect more episodes, add them to $\mathcal{D}$, and continue to update $\Theta$ from the augmented $\mathcal{D}$. This process repeats for multiple stages, and the model parameters learned from the previous stage will be used as the initialization for the next stage. Therefore, we can focus on learning at a single stage, which was formulated in Section 2 as one of solving the optimization problem (7). Note that the objective (7) is a function of the MAP estimates of $\theta_t^S$ and $\theta_t^{a_t}$. Therefore, we start with a recursion for computing $\hat{\theta}_t^S$ and $\hat{\theta}_t^{a_t}$ and then introduce our learning algorithm for $\Theta$.

### 3.1   Recursive MAP inference by mirror descent

The MAP estimates, $\hat{\theta}_t^S$ and $\hat{\theta}_t^a$, for the topic proportions $\theta_t^S$ and $\theta_t^a$ are defined as

$$(\hat{\theta}_t^S, \hat{\theta}_t^a) = \arg\max_{\theta_t^S, \theta_t^a} p(\theta_t^S, \theta_t^a | w_{1:t}^S, w_{1:t}^A, a_{1:t-1}) \tag{8}$$

Solving for the exact solution is, however, intractable. We instead develop an approximate algorithm that recursively estimate $\hat{\theta}_t^S$ and $\hat{\theta}_t^a$. To develop the algorithm, we rely on the following result, whose proof is deferred to Appendix A.

**Proposition 1.** *The MAP estimates in* (8) *could be approximated by recursively solving the problems:*

$$\hat{\theta}_t^S = \arg\max_{\theta_t^S} \left[ \ln p(x_t^S|\theta_t^S, \Phi_S) + \ln p(\theta_t^S|\hat{\alpha}_t^S) \right] \tag{9}$$

$$\hat{\theta}_t^a = \arg\max_{\theta_t^a} \left[ \ln p(x_t^a|\theta_t^a, \Phi_A) + \ln p(\theta_t^a|\hat{\alpha}_t^A) \right], \quad a \in \{1, \ldots, A_t\}, \tag{10}$$

*where $x_t^S$ and $x_t^a$ are the bag-of-words vectors for the observation text $w_t^S$ and the $a$-th action text $w_t^a$, respectively. To compute $\hat{\alpha}_t^S$ and $\hat{\alpha}_t^A$, we begin with $\hat{\alpha}_1^S = \alpha_1^S$ and $\hat{\alpha}_1^A = \alpha_1^A$ and update their values for the next $t + 1$ time step according to*

$$\hat{\alpha}_{t+1}^S = \sigma\left(W_{SS}\hat{\theta}_t^S + W_{SA}\hat{\theta}_t^{a_t} + \alpha_1^S\right), \quad \hat{\alpha}_{t+1}^A = \sigma\left(W_{AS}\hat{\theta}_t^S + W_{AA}\hat{\theta}_t^{a_t} + \alpha_1^A\right) \tag{11}$$

Note from (9)–(10) that, for given $\hat{\theta}_t^S$ and $\hat{\theta}_t^a$, the solution of $\theta_t^S$ and $\theta_t^a$ now becomes $A_t + 1$ decoupled sub-problems, each of which has the same form as the MAP inference problem of Chen et al. [8]. Therefore, we solve each sub-problem in (9)–(10) using their mirror descent inference algorithm, and then use (11) to compute the Dirichlet parameters at the next time step. The overall MAP inference procedure is summarized in Algorithm 2. We further remark that, after obtaining $\hat{\theta}_t^S$ and $\hat{\theta}_t^a$, the Q-value for the $t$ step is readily estimated by:

$$\mathbb{E}\left[Q(\theta_t^S, \theta_t^a)|w_{1:t}^S, w_{1:t}^A, a_{1:t-1}\right] \approx Q(\hat{\theta}_t^S, \hat{\theta}_t^a), \quad a \in \{1, \ldots, A_t\}, \tag{12}$$

where we approximate the conditional expectation using the MAP estimates. After learning is finished, the agent may extract a greedy policy for any state $s$ by taking the action $\arg\max_a Q(\hat{\theta}^S, \hat{\theta}^a)$. It is known that if the learned Q-function is closed to the true Q-function, such a greedy policy is near-optimal [21].

## 3.2 End-to-end learning by backpropagation

The training loss (7) for each learning stage has the form of a finite sum over $M$ episodes. Each term inside the summation depends on $\hat{\theta}_t^S$ and $\hat{\theta}_t^{a_t}$, which in turn depend on all the model parameters in $\Theta$ via the computational graph defined by Algorithm 2 (see Appendix E for a diagram of the graph). Therefore, we can learn the model parameters in $\Theta$ by sampling an episode in the data, computing the corresponding stochastic gradient in (7) by back-propagation on the computational graph given in Algorithm 2, and updating $\Theta$ by stochastic gradient/mirror descent. More details are found in Algorithm 1, and Appendix E.4 gives the gradient formulas.

## 4 Experiments

In this section, we use two text games from [11] to evaluate our proposed model and demonstrate the idea of interpreting the decision making processes: (i) "Saving John" and (ii) "Machine of Death" (see Appendix C for a brief introduction of the two games).[5] The action spaces of both games are defined by natural languages and the feasible actions change over time, which is a setting that Q-LDA is designed for. We choose to use the same experiment setup as [11] in order to have a fair comparison with their results. For example, at each $m$-th experience-replay learning (see Algorithm 1), we use the softmax action selection rule [21, pp.30–31] as the exploration policy to collect data (see Appendix E.3 for more details). We collect $M = 200$ episodes of data (about 3K time steps in "Saving John" and 16K in "Machine of Death") at each of $D = 20$ experience replays, which amounts to a total of $4,000$ episodes. At each experience replay, we update the model with 10 epochs before the next replay. Appendix E provides additional experimental details.

We first evaluate the performance of the proposed Q-LDA model by the long-term rewards it receives when applied to the two text games. Similar to [11], we repeat our experiments for five times with different random initializations. Table 1 summarize the means and standard deviations of the rewards

Table 1: The average rewards (higher is better) and standard deviations of different models on the two tasks. For DRRN and MA-DQN, the number of topics becomes the number of hidden units per layer.

| Tasks | # topics | Q-LDA | DRRN (1-layer) | DRRN (2-layer) | MA-DQN (2-layer) |
|-------|----------|-------|----------------|----------------|------------------|
| Saving John | 20 | **18.8** (0.3) | 17.1 (0.6) | 18.4 (0.1) | 4.9 (3.2) |
| | 50 | **18.6** (0.6) | 18.3 (0.2) | 18.5 (0.3) | 9.0 (3.2) |
| | 100 | **19.1** (0.6) | 18.2 (0.2) | 18.7 (0.4) | 7.1 (3.1) |
| Machine of Death | 20 | **19.9** (0.8) | 7.2 (1.5) | 9.2 (2.1) | 2.8 (0.9) |
| | 50 | **18.7** (2.1) | 8.4 (1.3) | 10.7 (2.7) | 4.3 (0.9) |
| | 100 | **17.5** (2.4) | 8.7 (0.9) | 11.2 (0.6) | 5.2 (1.2) |

on the two games. We include the results of Deep Reinforcement Relevance Network (DRRN) proposed in [11] with different hidden layers. In [11], there are several variants of DQN (deep Q-networks) baselines, among which MA-DQN (max-action DQN) is the best performing one. We therefore only include the results of MA-DQN. Table 1 shows that Q-LDA outperforms all other approaches on both tasks, especially "Machine of Death", where Q-LDA even beats the DRRN models by a large margin. The gain of Q-LDA on "Saving John" is smaller, as both Q-LDA and DRRN are approaching the upper bound of the reward, which is 20. "Machine of Death" was believed to be a more difficult task due to its stochastic nature and larger state and action spaces [11], where the upper bound on the reward is 30. (See Tables 4–5 for the definition of the rewards for different story endings.) Therefore, Q-LDA gets much closer to the upper bound than any other method, although there may still be room for improvement. Finally, our experiments follow the standard online RL setup: after a model is updated based on the data observed so far, it is tested on newly generated episodes. Therefore, the numbers reported in Table 1 are *not* evaluated on the training dataset, so they truthfully reflect the actual average reward of the learned models.

We now proceed to demonstrate the analysis of the latent pattern of the decision making process using one example episode of "Machine of Death". In this episode, the game starts with the player wandering in a shopping mall, after the peak hour ended. The player approaches a machine that prints a death card after inserting a coin. The death card hints on how the player will die in future. In one of the story development, the player's death is related to a man called Bon Jovi. The player is so scared that he tries to combat with a cardboard standee of Bon Jovi. He reveals his concern to a friend named Rachel, and with her help he finally overcomes his fear and maintains his friendship. This episode reaches a good ending and receives the highest possible reward of 30 in this game.

In Figure 2, we show the evolution of the topic proportions for the four most active topics (shown in Table 2)[6] for both the observation texts and the selected actions' texts. We note from Figure 2 that the most dominant observation topic and action topic at beginning of the episode are "wander at mall" and "action at mall", respectively, which is not surprising since the episode starts at a mall scenario. The topics related to "mall" quickly dies off after the player starts the death machine. Afterwards, the most salient observation topic becomes "meet Bon Jovi" and then "combat" ($t = 8$). This is because after the activation of death machine, the story enters a scenario where the player tries to combat with a cardboard standee. Towards the end of the episode, the observation topic "converse w/rachel" and the topic "kitchen & chat" corresponding to the selected action reach their peaks and then decay right before the end of the story, where the action topic "relieve" climbs up to its peak. This is consistent with the story ending, where the player chooses to overcome his fear after chatting with Rachel. In Appendix D, we show the observation and the action texts in the above stages of the story.

Finally, another interesting observation is about the matrix $U$. Since the Q-function value is computed from $[\hat{\theta}_t^a]^T U \hat{\theta}_t^S$, the $(i,j)$-th element of the matrix $U$ measures the positive/negative correlation between the $i$-th action topic and the $j$-th observation topic. In Figure 2(c), we show the value of the learned matrix $U$ for the four observation topics and the four action topics in Table 2. Interestingly, the largest value (39.5) of $U$ is the $(1, 2)$-th element, meaning that the action topic "relieve" and the state topic "converse w/rachel" has strong positive contribution to a high long-term reward, which is what happens at the end of the story.

Table 2: The four most active topics for the observation texts and the action texts, respectively.

| Observation Topics | |
|---|---|
| 1: combat | minutes, lights, firearm, shoulders, whiff, red, suddenly, huge, rendition |
| 2: converse w/ rachel | rachel, tonight, grabs, bar, towards, happy, believing, said, moonlight |
| 3: meet Bon Jovi | small, jovi, bon, door, next, dog, insists, room, wrapped, standees |
| 4: wander at mall | ended, catcher, shopping, peak, wrapped, hanging, attention, door |
| **Action Topics** | |
| 1: relieve | leave, get, gotta, go, hands, away, maybe, stay, ability, turn, easy, rachel |
| 2: kitchen & chat | wait, tea, look, brisk, classics, oysters, kitchen, turn, chair, moment |
| 3: operate the machine | coin, insert, west, cloth, desk, apply, dollars, saying, hands, touch, tell |
| 4: action at mall | alarm, machine, east, ignore, take, shot, oysters, win, gaze, bestowed |

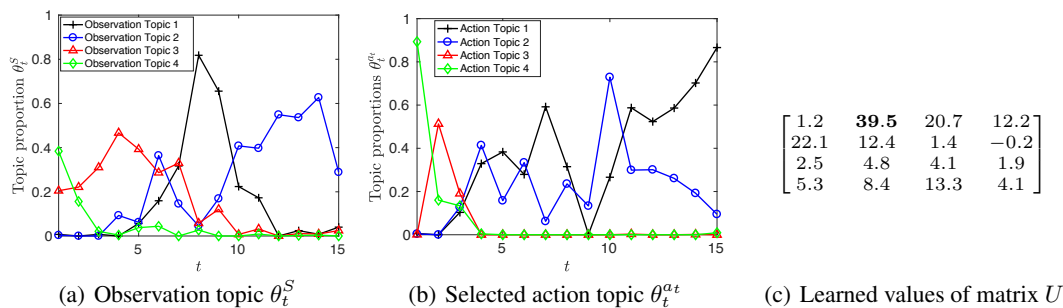

(a) Observation topic $\theta_t^S$  (b) Selected action topic $\theta_t^{a_t}$  (c) Learned values of matrix $U$

Figure 2: The evolution of the most active topics in "Machine of Death."

## 5   Conclusion

We proposed a probabilistic model, Q-LDA, to uncover latent patterns in text-based sequential decision processes. The model can be viewed as a latent topic model, which chains the topic proportions over time. Interestingly, by modeling the mean function of the immediate reward in a special way, we showed that discriminative learning of Q-LDA using its likelihood is closely related to Q-learning. Thus, our approach could also be viewed as a Q-learning variant for sequential topic models. We evaluate Q-LDA on two text-game tasks, demonstrating state-of-the-art rewards in these games. Furthermore, we showed our method provides a viable approach to finding interesting latent patterns in such decision processes.

### Acknowledgments

The authors would like to thank all the anonymous reviewers for their constructive feedback.

## Footnotes

[2]For notation simplicity, we assume all texts have the same length $N$.

[4]The intuition of choosing $Q(\cdot, \cdot)$ to be this form is that we want $\theta_t^S$ to be aligned with $\theta_t^a$ of the correct action (large Q-value), and to be misaligned with the $\theta_t^a$ of the wrong actions (small Q-value). The introduction of $U$ allows the number and the meaning of topics for the observations and actions to be different.

[5]The simulators are obtained from https://github.com/jvking/text-games

[6]In practice, we observe that some topics are never or rarely activated during the learning process. This is especially true when the number of topics becomes large (e.g., 100). Therefore, we only show the most active topics. This might also explain why the performance improvement is marginal when the number of topics grows.

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
