[Supplementary Material · NIPS2017_ReinforcementLDA_SupplementaryMaterial.pdf]

# Supplementary Material for "Q-LDA: Uncovering Latent Patterns in Text-based Sequential Decision Processes"

## A Proof of Proposition 1

We first write out the joint probability of the Q-LDA model as

$$
\prod_{i=1}^{M}\prod_{t=0}^{T_i} \left\{ \pi_b(a_t|w_{1:t}^S, w_{1:t}^A, a_{1:t-1}) \times p(w_t^S, z_t^S, \theta_t^S|\alpha_t^S, \Phi_S) p(\alpha_t^S|\theta_{t-1}^S, \theta_{t-1}^{a_{t-1}}, W_{SS}, W_{SA}) p(\Phi_S|\beta_S) \right.
$$

$$
\left. \times p(w_t^A, z_t^A, \theta_t^A|\alpha_t^A, \Phi_A) p(\alpha_t^A|\theta_{t-1}^S, \theta_{t-1}^{a_{t-1}}, W_{AS}, W_{AA}) p(\Phi_A|\beta_A) \times p(r_t|\theta_t^S, \theta_t^{a_t}, U) \right\} \quad (13)
$$

where $\theta_t^A \triangleq \{\theta_t^a\}$, $z_t^S \triangleq \{z_{t,n}^S\}$ and $z_t^A \triangleq \{z_{t,n}^a\}$. Following the same line of argument as in [8], we marginalize the variables $z_t^S$ and $z_t^A$ in joint probability of the Q-LDA model and obtain

$$
\prod_{i=1}^{M}\prod_{t=0}^{T_i} \left\{ \pi_b(a_t|w_{1:t}^S, w_{1:t}^A, a_{1:t-1}) \times p(x_t^S, \theta_t^S|\alpha_t^S, \Phi_S) p(\alpha_t^S|\theta_{t-1}^S, \theta_{t-1}^{a_{t-1}}, W_{SS}, W_{SA}) p(\Phi_S|\beta_S) \right.
$$

$$
\left. \times p(x_t^A, \theta_t^A|\alpha_t^A, \Phi_A) p(\alpha_t^A|\theta_{t-1}^S, \theta_{t-1}^{a_{t-1}}, W_{AS}, W_{AA}) p(\Phi_A|\beta_A) \times p(r_t|\theta_t^S, \theta_t^{a_t}, U) \right\} \quad (14)
$$

where $x_t^S$ is the bag-of-words (BOW) vector for the observation text at the $t$-th time step. Note that the probability depends on $w_{1:t}^S$ and $w_{1:t}^A$ via $x_{1:t}^S$ and $x_{1:t}^A$. Therefore, we can also write the policy as $\pi_b(a_t|x_{1:t}^S, x_{1:t}^A, a_{1:t-1})$ so that

$$
\prod_{i=1}^{M}\prod_{t=0}^{T_i} \left\{ \pi_b(a_t|x_{1:t}^S, x_{1:t}^A, a_{1:t-1}) \times p(x_t^S, \theta_t^S|\alpha_t^S, \Phi_S) p(\alpha_t^S|\theta_{t-1}^S, \theta_{t-1}^{a_{t-1}}, W_{SS}, W_{SA}) p(\Phi_S|\beta_S) \right.
$$

$$
\left. \times p(x_t^A, \theta_t^A|\alpha_t^A, \Phi_A) p(\alpha_t^A|\theta_{t-1}^S, \theta_{t-1}^{a_{t-1}}, W_{AS}, W_{AA}) p(\Phi_A|\beta_A) \times p(r_t|\theta_t^S, \theta_t^{a_t}, U) \right\} \quad (15)
$$

First, by Bayes rule, we have

$$
p(\theta_t^S, \theta_t^A|x_{1:t}^S, x_{1:t}^A, a_{1:t-1}) = \frac{p(\theta_t^S, \theta_t^A, x_t^S, x_t^A|x_{1:t-1}^S, x_{1:t-1}^A, a_{1:t-1})}{p(x_t^S, x_t^A|x_{1:t-1}^S, x_{1:t-1}^A, a_{1:t-1})} \quad (16)
$$

where for simplicity of notation we dropped the dependency on the model parameters $\Theta \triangleq (\Phi_S, \Phi_A, W_{SS}, W_{SA}, W_{AS}, W_{AA}, U)$. Note that the denominator is independent of $(\theta_t^S, \theta_t^A)$. Therefore, the MAP estimate of $(\theta_t^S, \theta_t^A)$ is the same as maximizing the numerator:

$$
(\hat{\theta}_t^S, \hat{\theta}_t^A) \triangleq \arg\max_{\theta_t} p(\theta_t^S, \theta_t^A, x_t^S, x_t^A|x_{1:t-1}^S, x_{1:t-1}^A, a_{1:t-1}) \quad (17)
$$

We now proceed to compute the probability $p(\theta_t^S, \theta_t^A, x_t^S, x_t^A|x_{1:t-1}^S, x_{1:t-1}^A, a_{1:t-1})$. Note that

$$
p(\theta_t^S, \theta_t^A, x_t^S, x_t^A|x_{1:t-1}^S, x_{1:t-1}^A, a_{1:t-1})
$$
$$
= \int p(x_t^S|\theta_t^S, \Phi_S) p(\theta_t^S|\alpha_t^S) p(\alpha_t^S|\theta_{t-1}^S, \theta_{t-1}^{a_{t-1}}, W_{SS}, W_{SA})
$$
$$
\times p(x_t^A|\theta_t^A, \Phi_A) p(\theta_t^A|\alpha_t^A) p(\alpha_t^A|\theta_{t-1}^S, \theta_{t-1}^{a_{t-1}}, W_{AA}, W_{AS})
$$
$$
\times p(\theta_{t-1}^S, \theta_{t-1}^A|x_{1:t-1}^S, x_{1:t-1}^A, a_{1:t-1}) d\alpha_t^S d\alpha_t^A d\theta_{t-1}^S d\theta_{t-1}^A \quad (18)
$$

Note that the random variable $a_{t-1}$ is generated according to $\pi_b(a_{t-1}|x_{1:t-1}^S, x_{1:t-1}^A, a_{1:t-2})$, which is conditioned on $x_{1:t-1}^S$, $x_{1:t-1}^A$ and $a_{1:t-2}$. Therefore, knowing $a_{t-1}$ does not provide additional information regarding $\theta_{t-1}^S$ and $\theta_{t-1}^A$ once $x_{1:t-1}^S$, $x_{1:t-1}^A$ and $a_{1:t-2}$ are known, which leads to the following relation:

$$
p(\theta_{t-1}^S, \theta_{t-1}^A|x_{1:t-1}^S, x_{1:t-1}^A, a_{1:t-1}) = p(\theta_{t-1}^S, \theta_{t-1}^A|x_{1:t-1}^S, x_{1:t-1}^A, a_{1:t-2}) \quad (19)
$$

Substituting the above expression into (18), we obtain

$$p(\theta_t^S, \theta_t^A, x_t^S, x_t^A | x_{1:t-1}^S, x_{1:t-1}^A, a_{1:t-1})$$

$$= \int p(x_t^S | \theta_t^S, \Phi_S) p(\theta_t^S | \alpha_t^S) p(\alpha_t^S | \theta_{t-1}^S, \theta_{t-1}^{a_{t-1}}, W_{SS}, W_{SA})$$

$$\times\, p(x_t^A | \theta_t^A, \Phi_A) p(\theta_t^A | \alpha_t^A) p(\alpha_t^A | \theta_{t-1}^S, \theta_{t-1}^{a_{t-1}}, W_{AS}, W_{AA})$$

$$\times\, p(\theta_{t-1}^S, \theta_{t-1}^A | x_{1:t-1}^S, x_{1:t-1}^A, a_{1:t-1}) d\alpha_t^S d\alpha_t^A d\theta_{t-1}^S d\theta_{t-1}^A$$

$$= \int p(x_t^S | \theta_t^S, \Phi_S) p(\theta_t^S | \alpha_t^S) p(\alpha_t^S | \theta_{t-1}^S, \theta_{t-1}^{a_{t-1}}, W_{SS}, W_{SA})$$

$$\times\, p(x_t^A | \theta_t^A, \Phi_A) p(\theta_t^A | \alpha_t^A) p(\alpha_t^A | \theta_{t-1}^S, \theta_{t-1}^{a_{t-1}}, W_{AS}, W_{AA})$$

$$\times\, p(\theta_{t-1}^S, \theta_{t-1}^A | x_{1:t-1}^S, x_{1:t-1}^A, a_{1:t-2}) d\alpha_t^S d\alpha_t^A d\theta_{t-1}^S d\theta_{t-1}^A$$

$$\overset{(a)}{=} \int p(x_t^S | \theta_t^S, \Phi_S) p(\theta_t^S | \alpha^S(\theta_{t-1}^S, \theta_{t-1}^{a_{t-1}}, W_{SS}, W_{SA}))$$

$$\times\, p(x_t^A | \theta_t^A, \Phi_A) p(\theta_t^A | \alpha^A(\theta_{t-1}^S, \theta_{t-1}^{a_{t-1}}, W_{AS}, W_{AA}))$$

$$\times\, p(\theta_{t-1}^S, \theta_{t-1}^A | x_{1:t-1}^S, x_{1:t-1}^A, a_{1:t-2}) d\theta_{t-1}^S d\theta_{t-1}^A$$

$$\overset{(b)}{\approx} p(x_t^S | \theta_t^S, \Phi_S) p(\theta_t^S | \alpha^S(\hat{\theta}_{t-1}^S, \hat{\theta}_{t-1}^{a_{t-1}}, W_{SS}, W_{SA}))$$

$$\times\, p(x_t^A | \theta_t^A, \Phi_A) p(\theta_t^A | \alpha^A(\hat{\theta}_{t-1}^S, \hat{\theta}_{t-1}^{a_{t-1}}, W_{AA}, W_{AS})) \tag{20}$$

where step (a) uses the fact that the probability distribution of $\alpha_t^S$ and $\alpha_t^A$ are Dirac delta functions and step (b) samples the integral with MAP estimates of $\theta_t^S$ and $\theta_t^A$. Therefore, substituting (20) into (17), we get

$$(\hat{\theta}_t^S, \hat{\theta}_t^A) \approx \arg\max_{(\theta_t^S, \theta_t^A)} \left\{ p(x_t^S | \theta_t^S, \Phi_S) p(\theta_t^S | \alpha^S(\hat{\theta}_{t-1}^S, \hat{\theta}_{t-1}^{a_{t-1}}, W_{SS}, W_{SA})) \right.$$

$$\left. \times\, p(x_t^A | \theta_t^A, \Phi_A) p(\theta_t^A | \alpha^A(\hat{\theta}_{t-1}^S, \hat{\theta}_{t-1}^{a_{t-1}}, W_{AS}, W_{AA})) \right\}$$

$$= \arg\max_{(\theta_t^S, \theta_t^A)} \left\{ \ln p(x_t^S | \theta_t^S, \Phi_S) + \ln p(\theta_t^S | \alpha^S(\hat{\theta}_{t-1}^S, \hat{\theta}_{t-1}^{a_{t-1}}, W_{SS}, W_{SA})) \right.$$

$$\left. + \ln p(x_t^A | \theta_t^A, \Phi_A) + \ln p(\theta_t^A | \alpha^A(\hat{\theta}_{t-1}^S, \hat{\theta}_{t-1}^{a_{t-1}}, W_{AS}, W_{AA})) \right\} \tag{21}$$

Using the definition of these probability distributions, we can show that the above MAP estimation problem can be decomposed into

$$\hat{\theta}_t^S = \arg\max_{\theta_t^S} \left[ \ln p(x_t^S | \theta_t^S, \Phi_S) + \ln p(\theta_t^S | \alpha^S(\hat{\theta}_{t-1}^S, \hat{\theta}_{t-1}^{a_{t-1}}, W_{SS}, W_{SA})) \right] \tag{22}$$

$$\hat{\theta}_t^a = \arg\max_{\theta_t^a} \left[ \ln p(x_t^a | \theta_t^a, \Phi_A) + \ln p(\theta_t^a | \alpha^A(\hat{\theta}_{t-1}^S, \hat{\theta}_{t-1}^{a_{t-1}}, W_{AS}, W_{AA})) \right]$$

$$a = 1, \ldots, A_t \tag{23}$$

Note that the approximate MAP inference of $\theta_t^S$ and $\theta_t^a$ $(a = 1, \ldots, A_t)$ is completely decoupled into independent optimization problems, which could be solved by mirror descent separately. Therefore, we complete our proof of Proposition 1.

## B  Approximation of the learning objective function

In this appendix, we show that the learning objective function (3) can be approximated by the cost function (7). For convenience, we repeat (3) below:

$$\max_{\Theta} \left\{ \ln p(\Theta) + \sum_{i=1}^{M} \ln p(r_{1:T_i} | x_{1:T_i}^S, x_{1:T_i}^A, a_{1:T_i}, \Theta) \right\} \tag{24}$$

An important step of our derivation is to write $p(r_{1:T_i} | x_{1:T_i}^S, x_{1:T_i}^A, a_{1:T_i}, \Theta)$ as an expression of probabilities for each time step $t$. We begin by examining the joint probability $p(x_{1:T_i}^S, x_{1:T_i}^A a_{1:T_i}, r_{1:T_i} | \Theta)$:

$$p(x_{1:T_i}^S, x_{1:T_i}^A, a_{1:T_i}, r_{1:T_i} | \Theta)$$

$$= \prod_{t=1}^{T_i} p(x_t^S, x_t^A, a_t, r_t | x_{1:t-1}^S, x_{1:t-1}^A, a_{1:t-1}, r_{1:t-1}, \Theta)$$

$$= \prod_{t=1}^{T_i} p(x_t^S, x_t^A | x_{1:t-1}^S, x_{1:t-1}^A, a_{1:t-1}, r_{1:t-1}, \Theta) \times \pi(a_t | x_{1:t}^S, x_{1:t}^A, a_{1:t-1}, r_{1:t-1})$$

$$\times p(r_t | x_{1:t}^S, x_{1:t}^A, a_{1:t}, r_{1:t-1}, \Theta)$$

$$= \prod_{t=1}^{T_i} p(x_t^S, x_t^A | x_{1:t-1}^S, x_{1:t-1}^A, a_{1:t-1}, \Theta) \pi(a_t | x_{1:t}^S, x_{1:t}^A, a_{1:t-1}) p(r_t | x_{1:t}^S, x_{1:t}^A, a_{1:t}, \Theta) \quad (25)$$

where the last step uses the fact that the behavior policy for exploring the environment does not depend on the current model parameter to be optimized and the fact that the intermediate rewards are known deterministic quantities except the terminal reward. Likewise, we can also get

$$p(x_{1:T_i}^S, x_{1:T_i}^A, a_{1:T_i} | \Theta) = \prod_{t=1}^{T_i} p(x_t^S, x_t^A, a_t | x_{1:t-1}^S, x_{1:t-1}^A, a_{1:t-1}, \Theta)$$

$$= \prod_{t=1}^{T_i} p(x_t^S, x_t^A | x_{1:t-1}^S, x_{1:t-1}^A, a_{1:t-1}, \Theta) \pi(a_t | x_{1:t}^S, x_{1:t}^A, a_{1:t-1}) \quad (26)$$

Dividing (25) by the above expression leads to

$$p(r_{1:T_i} | x_{1:T_i}^S, x_{1:T_i}^A, a_{1:T_i}, \Theta) = \frac{p(x_{1:T_i}^S, x_{1:T_i}^A, a_{1:T_i}, r_{1:T_i} | \Theta)}{p(x_{1:T_i}^S, x_{1:T_i}^A, a_{1:T_i} | \Theta)} = \prod_{t=1}^{T_i} p(r_t | x_{1:t}^S, x_{1:t}^A, a_{1:t}, \Theta) \quad (27)$$

We now examine the term inside the product of (27). Unfortunately, the exact expression is not tractable as it requires to marginalize out all the latent variables, which cannot be done in closed-form. Instead, we develop approximate expressions for it. Note that

$$p(r_t | x_{1:t}^S, x_{1:t}^A, a_{1:t}, \Theta) = \int p(r_t | \theta_t^S, \theta_t^{a_t}, U) p(\theta_t^S, \theta_t^A | x_{1:t}^S, x_{1:t}^A, a_{1:t}, \Theta) d\theta_t^S d\theta_t^A$$

$$\overset{(a)}{=} \int p(r_t | \theta_t^S, \theta_t^{a_t}, U) p(\theta_t^S, \theta_t^A | x_{1:t}^S, x_{1:t}^A, a_{1:t-1}, \Theta) d\theta_t^S d\theta_t^A$$

$$= \mathbb{E}_{\theta_t^S, \theta_t^{a_t} | x_{1:t}^S, x_{1:t}^A, a_{1:t-1}} \left[ p(r_t | \theta_t^S, \theta_t^{a_t}, U) \right]$$

$$\overset{(b)}{\approx} p(r_t | \hat{\theta}_t^S, \hat{\theta}_t^{a_t}, U) \quad (28)$$

where step (a) uses the fact that the action $a_t$ is generated only by $x_{1:t}^S$, $x_{1:t}^A$ and $a_{1:t-1}$, and step (b) approximate the expectation by sampling it with the MAP estimate. Substituting (28) into (27), we get

$$p(r_{1:T_i} | x_{1:T_i}^S, x_{1:T_i}^A, a_{1:T_i}, \Theta) = \prod_{t=1}^{T_i} \mathbb{E}_{\theta_t^S, \theta_t^{a_t} | x_{1:t}^S, x_{1:t}^A, a_{1:t-1}} \left[ p(r_t | \theta_t^S, \theta_t^{a_t}, U) \right] \approx \prod_{t=1}^{T_i} p(r_t | \hat{\theta}_t^S, \hat{\theta}_t^{a_t}, U) \quad (29)$$

Substituting (29) into (24), we obtain

$$\max_{\Theta} \left\{ \ln p(\Theta) + \sum_{i=1}^{M} \sum_{t=1}^{T_i} \ln p(r_t | \hat{\theta}_t^S, \hat{\theta}_t^{a_t}, U) \right\} \quad (30)$$

Recalling from (1) that, conditioned on $\hat{\theta}_t^S$ and $\hat{\theta}_t^{a_t}$, $r_t$ is a Gaussian random variable with mean $\mu_r(\hat{\theta}_t^S, \hat{\theta}_t^{a_t}, U)$ and variance $\sigma_r^2$, we can express $p(r_t | \hat{\theta}_t^S, \hat{\theta}_t^{a_t}, U)$ as:

$$p(r_t | \hat{\theta}_t^S, \hat{\theta}_t^{a_t}, U) = \frac{1}{\sqrt{2\pi\sigma_r^2}} \exp\left( -\frac{1}{2\sigma_r^2} (r_t - \mu_r(\hat{\theta}_t^S, \hat{\theta}_t^{a_t}, U))^2 \right)$$

$$\overset{(a)}{=} \frac{1}{\sqrt{2\pi\sigma_r^2}} \exp\left( -\frac{1}{2\sigma_r^2} (r_t - Q(\hat{\theta}_t^S, \hat{\theta}_t^{a_t}) + \gamma \cdot \mathbb{E}\left[ \max_{a_{t+1}} Q(\theta_{t+1}^S, \theta_{t+1}^{a_{t+1}}) | \hat{\theta}_t^S, \hat{\theta}_t^{a_t} \right])^2 \right)$$

$$(31)$$

where step (a) substituted (6). Substituting (31) into (30), we obtain

$$\min_{\Theta} \left\{ - \ln p(\Theta) + \sum_{i=1}^{M} \sum_{t=1}^{T_i} \frac{1}{2\sigma_r^2} \left\| r_t - Q(\hat{\theta}_t^S, \hat{\theta}_t^{a_t}) + \gamma \cdot \mathbb{E} \left[ \max_{a_{t+1}} Q(\theta_{t+1}^S, \theta_{t+1}^{a_{t+1}}) | \hat{\theta}_t^S, \hat{\theta}_t^{a_t} \right] \right\|^2 \right\} \tag{32}$$

where we have dropped some constant terms. Introduce

$$d_t = \begin{cases} r_t + \gamma \cdot \mathbb{E}_{\theta_{t+1}^S, \theta_{t+1}^b | \hat{\theta}_t^S, \hat{\theta}_t^{a_t}} [\max_b Q(\theta_{t+1}^S, \theta_{t+1}^b)] & t < T_i \\ r_{T_i} & t = T_i \end{cases} \tag{33}$$

Then we can write (32) as

$$\min_{\Theta} \left\{ - \ln p(\Theta) + \sum_{i=1}^{M} \sum_{t=1}^{T_i} \frac{1}{2\sigma_r^2} \left[ d_t - Q(\hat{\theta}_t^S, \hat{\theta}_t^{a_t}) \right]^2 \right\} \tag{34}$$

A remaining problem is that $d_t$ has a conditional expectation with respect to $\theta_{t+1}^S$ and $\theta_{t+1}^{a_{t+1}}$. First, note that we can have the following approximation:

$$\mathbb{E}_{\theta_t^S, \theta_t^{a_t} | x_{1:t}^S, x_{1:t}^A, a_{1:t-1}} \left\{ \mathbb{E}_{\theta_{t+1}^S, \theta_{t+1}^b | \theta_t^S, \theta_t^{a_t}} [\max_b Q(\theta_{t+1}^S, \theta_{t+1}^b)] \right\} \approx \mathbb{E}_{\theta_{t+1}^S, \theta_{t+1}^b | \hat{\theta}_t^S, \hat{\theta}_t^{a_t}} [\max_b Q(\theta_{t+1}^S, \theta_{t+1}^b)]$$

where we sample the outer conditional expectation by the MAP samples $\hat{\theta}_t^S$ and $\hat{\theta}_t^{a_t}$. Then, we have

$$\mathbb{E}_{\theta_{t+1}^S, \theta_{t+1}^b | \hat{\theta}_t^S, \hat{\theta}_t^{a_t}} [\max_b Q(\theta_{t+1}^S, \theta_{t+1}^b)]$$

$$\approx \mathbb{E}_{\theta_t^S, \theta_t^{a_t} | x_{1:t}^S, x_{1:t}^A, a_{1:t-1}} \left\{ \mathbb{E}_{\theta_{t+1}^S, \theta_{t+1}^b | \theta_t^S, \theta_t^{a_t}} [\max_b Q(\theta_{t+1}^S, \theta_{t+1}^b)] \right\}$$

$$= \mathbb{E}_{\theta_{t+1}^S, \theta_{t+1}^b | x_{1:t}^S, x_{1:t}^A, a_{1:t-1}} \left\{ \max_b Q(\theta_{t+1}^S, \theta_{t+1}^b) \right\}$$

$$\approx \max_b Q(\hat{\theta}_{t+1}^S, \hat{\theta}_{t+1}^b) \tag{35}$$

In summary, we have the approximation:

$$d_t = \begin{cases} r_t + \gamma \max_b Q(\hat{\theta}_{t+1}^S, \hat{\theta}_{t+1}^b) & t < T_i \\ r_t & t = T_i \end{cases} \tag{36}$$

which completes our proof.

## C   Introduction of the two text games

In Figure 3, we show two screenshots of the two text games used in this paper. The first game belongs to choice-based game, where the feasible actions at each time are listed separately as candidate choices. And the second game is a mix between choice-based and hypertext-based game (where the actions are embedded in the observation text as substrings with hyperlinks). The action spaces of both games are defined by natural languages and the feasible actions change over time, which is a setting that Q-LDA is designed for. This setting was believed to be more challenging than the parser-based text games in [19], which accepts a (small) fixed set of pre-defined typed-in commands (e.g., "eat apple", "get key"). Therefore, we do not consider parser-based game and will focus on the choice-based and hyperlink-based games. To be self-contained, we include the description of the two text games ("Saving John" and "Machine of Death") from [11] (Tables 3, 4, and 5). Table 3 gives the basic statistics of the two text games, Tables 4-5 give the rewards for different endings of the two games. In Table 6, we give an example text flow when playing "Machine of Death". In addition, the number of conversation turns (number of steps) per episode is 10-30 for "SavingJohn" and is 10-200 for "Machine of Death". When the training converges, the length is around 7 for "SavingJohn" and is between 10-20 for "Machine of Death". For more details, the readers are referred to [11] and its supplementary material.

"Save me? She couldn't even save herself if she tried."

I remember Adam telling me.

"Like the time we were on that project together,"

He never could let that go.

"Bitch is neuro-psycho man, I dunno what you see in her. Honest to God, people really shouldn't work when they're sick. And Cherie's one sick pussy!"

Adam's always known how to push my buttons. As I focus on the memories of Adam, I could feel myself stiffening up.

**Keep thinking about Adam**   **Forget about Adam**   **Focus on another memory**

You explain your dire situation to the old man, who reveals his name to be John.

"Alright," he says with a sniff. "As for the axe, I was out choppin' firewood. I'm not sure if you've noticed, but it's bloody cold."

"The storm knocked out the phone line. The weather's died down enough that the CB Radio in my truck might work."

He opens the door you crashed through earlier, and before closing it behind him, looks you in the eye and says "Stay right there. And don't touch anything."

You look around the room you're in and find it to be a small, modest kitchen. There's a sink with a **kettle** to its left, a collection of **drawers** to its right, and a **cupboard** below it. A **pantry** door stands tall in the corner, with a number of **framed photographs** displayed on the wall next to it. A **table** sits at the centre of it all.

Stairs lead **up** and a door leads **east**. Another door leads back **outside**.

The **axe** remains stuck in the floor.

**Wait.**

(a) "Saving John"      (b) "Machine of Death"

Figure 3: The user interface of the two text games used for evaluation.

Table 3: Statistics for the games "Saving John" and "Machine of Death".

| Game | Saving John | Machine of Death |
|---|---|---|
| Text game type | Choice | Choice & Hypertext |
| Vocab size | 1762 | 2258 |
| Action vocab size | 171 | 419 |
| Avg. words/description | 76.67 | 67.80 |
| State transitions | Deterministic | Stochastic |
| # of states (underlying) | $\geq 70$ | $\geq 200$ |

# D  Additional experiment results

In Table 6, we show the snapshots of the text game "Machine of Death" at three different time steps: beginning ($t = 2$), in the middle ($t = 8$), and approaching the end ($t = 15$). In the table, we show the observation texts and the action texts for all the actions. The action texts highlighted in boldface correspond to the selected action. Below, we show the value of the matrix $U$ in the learned model parameter on "Machine of Death" task:

$$U = \begin{bmatrix} 1.2014 & \mathbf{39.5233} & 20.7054 & 12.2296 \\ 22.1366 & 12.4041 & 1.3726 & -0.1604 \\ 2.5195 & 4.8452 & 4.1210 & 1.9419 \\ 5.3332 & 8.3989 & 13.3208 & 4.1159 \end{bmatrix} \tag{37}$$

# E  Implementation details

## E.1  Details of the inference algorithm

As we discussed in the paper, we use mirror descent algorithm to perform MAP inference. In Algorithm 2, the MAP inference is implemented with constant step-size $\delta$. In practice, we found that it converges faster if we use adaptive step-size determined by line search. In Algorithm 3, we include the mirror descent inference algorithm with line search.

## E.2  Details of the learning algorithm

In Figure 4, we visualize the computation graph of the inference step time $t$, which illustrates the recursive inference steps in Algorithm 2 (or Algorithm 3). We observe that the recursive inference process could be interpreted as a recurrent neural network (RNN) with the following special structures. The topic distributions $\theta_t^S$ and $\{\theta_t^a\}$ can be viewed as $A_t + 1$ (time-varying) sets of hidden units that satisfy probabilistic simplex constraints, which are computed by $A_t + 1$ feedforward mirror descent networks (Figure 4(b)) from the input vectors $x_t^S$ and $\{x_t^a\}$ and the Dirichlet parameters $\hat{\alpha}_t^S$ and $\hat{\alpha}_t^A$. The recurrent links from the current hidden units ($\theta_t^S$ and $\{\theta_t^A\}$) to the next ones are through

Table 4: Final rewards defined for the text game "Saving John"

| Reward | Endings (partially shown) |
|---|---|
| -20 | Suspicion fills my heart and I scream. Is she trying to kill me? I don't trust her one bit... |
| -10 | Submerged under water once more, I lose all focus... |
| 0 | Even now, she's there for me. And I have done nothing for her... |
| 10 | Honest to God, I don't know what I see in her. Looking around, the situation's not so bad... |
| 20 | Suddenly I can see the sky... I focus on the most important thing - that I'm happy to be alive. |

Table 5: Final rewards for the text game "Machine of Death." Scores are assigned according to whether the character survives, how the friendship develops, and whether he overcomes his fear.

| Reward | Endings (partially shown) |
|---|---|
| -20 | You spend your last few moments on Earth lying there, shot through the heart, by the image of Jon Bon Jovi. |
| -20 | you hear Bon Jovi say as the world fades around you. |
| -20 | As the screams you hear around you slowly fade and your vision begins to blur, you look at the words which ended your life. |
| -10 | You may be locked away for some time. |
| -10 | Eventually you're escorted into the back of a police car as Rachel looks on in horror. |
| -10 | Fate can wait. |
| -10 | Sadly, you're so distracted with looking up the number that you don't notice the large truck speeding down the street. |
| -10 | All these hiccups lead to one grand disaster. |
| 10 | Stay the hell away from me! She blurts as she disappears into the crowd emerging from the bar. |
| 20 | You can't help but smile. |
| 20 | Hope you have a good life. |
| 20 | Congratulations! |
| 20 | Rachel waves goodbye as you begin the long drive home. After a few minutes, you turn the radio on to break the silence. |
| 30 | After all, it's your life. It's now or never. You ain't gonna live forever. You just want to live while you're alive. |

the Dirichlet parameters computed via (11). Furthermore, there are $A_t + 1$ output units, which are pairwise bilinear functions of $\theta_t^S$ and $\theta_t^a$ for each $a = 1, \ldots, A_t$. Therefore, the entire inference process could be interpreted as using a special structured RNN to approximate the Q-function in reinforcement learning. From this perspective, our work is related to DRRN [11] in that both of them use separate embedding vectors for the state and action texts and that they both use bilinear functions to map the embeddings into a Q-value. However, our work uses a special structured RNN to embed the input texts into their respective representation vectors while DRRN uses standard feedforward DNN. Our work is also related to the deep recurrent Q-network (DRQN) [10], which uses standard RNN (rather than the special structured RNN in our case) to approximate the Q-function to address the partial observability problem in reinforcement learning. Different from our model, the DRQN only works in the case with a fixed action space and could not handle the situation where the actions are described by natural languages. Finally, the above special RNN structures are designed from the generative model of Q-LDA, while both DRRN and DRQN are constructed as a black-box model for function approximation in Q-learning. This enables Q-LDAto be more interpretable during the decision making process.

### E.3    Details of the experiments

The softmax action selection rule for behavior policy can be written as $\pi_b^m(a_t|x_{1:t}^S, x_{1:t}^A, a_{1:t-1}) \propto \exp[\frac{1}{\tau} Q(\hat{\theta}_t^S, \hat{\theta}_t^a)]$ for all $a_t = 1, \ldots, A_t$, where $\tau$ is a temperature parameter that controls the sharpness of the softmax. $Q(\hat{\theta}_t^S, \hat{\theta}_t^a)$ is computed according to Algorithm 2 using the model parameter $\Theta_{m-1}$ from the previous experience replay. That is, at the exploration stage of each $m$-th experience replay, the behavior policy $\pi_b^m(\cdot)$ is parameterized by $\Theta_{m-1}$, which will be fixed during the exploration stage. With this, the behavior policy $\pi_b^m$ can be viewed as independent of the model parameter $\Theta$ to be optimized in the $m$-th replay. During the exploration stage, we will terminate the episode

Table 6: Snapshots of game observation and actions at different times for "Machine of Death"

| Time step | $t = 2$ | $t = 8$ | $t = 15$ |
|---|---|---|---|
| Observation text (partially shown) | You approach The Machine, which has the very charming street name of The Machine of Death. The device has only been around for a few years, but it's already hard to imagine a world without it, as it completely reshaped it, creating a culture of death. ... You never did get yourself tested. Maybe today is the day. | You decided that you don't need a firearm. You already have a set of guns sitting below your shoulders, after all. ... You take a moment to relish the drunken merriment. Then, in a corner, you see rock idol Jon Bon Jovi. | 'It makes me feel normal,' she admits. ... she says with a laugh. 'I'm going to go let Bonny have a run. You better be careful around him,' she adds with a mischievous grin. ...... People call Rachel the crazy one, but you're the one carrying a gun around in case you bump into members of Bon Jovi! |
| Action texts (selected action in **bold**) | [1] Return your eyes to the mall. [2] A slip of paper is stuck to the side of the Machine. Examine it. [3] Stand back and watch people use the Machine. **[4] Insert a coin.** | **[1] Duck! DUCK!** [2] Tackle him to the ground! [3] Ignore him. | **[1] It's time to let it go. Dismantle the gun to the best of your ability and get rid of it.** [2] Things could have gone a lot worse tonight. Who knows when I'll need that gun to survive! |

(a) The computational graph at each time step $t$
(b) The computation graph of mirror descent

Figure 4: (a) Feedforward computation graph for the model in Figure 1. We use the same blue color for the mirror descent graphs on the action texts to represent that they share the same model parameter $\Phi_A$. The mirror descent graph for the state text uses a different yellow color to imply that it uses a different model parameter $\Phi_S$. (b) The mirror descent graph in (a), where $\Phi$ is either $\Phi_S$ or $\Phi_A$.

when its length exceeds 100 in "Saving John" and will terminate the episode when the length exceeds 500 in "Machine of Death".

For learning algorithm, we use RMSProp to adaptively adjust the learning rate for each model parameter, with exponential decaying parameter $0.999$. The overall learning rates are chosen to be:

- $\mu_U = 1.0$ for both games
- In "Saving John", $\mu_{\Phi_S} = \mu_{\Phi_A} = 10^{-4}$ when the number of topics is 20 and 50, and $10^{-5}$ when the number of topics is 100. In "Machine of Death", $\mu_{\Phi_S} = \mu_{\Phi_A} = 10^{-4}$ when the number of topics is 20 and 50, and $10^{-6}$ when the number of topics is 100.
- In "Saving John", the learning rates for all $W_{SS}, W_{SA}, W_{AS}, W_{AA}$ are chosen to be $10^{-2}$. In "Machine of Death", they are chosen to be $10^{-2}$ when the number of topics is 20 and 50, and $10^{-3}$ when the number of topics is 100.

**Algorithm 3** Inference with Mirror-Descent over one episode (with line search)

1: **for** $t = 0, \ldots, T_i$ **do**
2:  **if** $t = 0$ **then**
3:    $\hat{\alpha}_0^S = \alpha_0^S$ and $\hat{\alpha}_0^A = \alpha_0^A$.
4:  **else**
5:    $\hat{\alpha}_t^S = \alpha^S(\theta_{t-1,L}^S, \theta_{t-1,L}^{a_{t-1}}, W_S)$ and $\hat{\alpha}_t^A = \alpha^A(\theta_{t-1,L}^S, \theta_{t-1,L}^{a_{t-1}}, W_A)$
6:  **end if**
7:  Initialization: $\theta_{t,0}^S = \frac{1}{K}\mathbb{1}$ and $T_{t,0}$.
8:  **for** $\ell = 1, \ldots, L$ **do**
9:    $T_{t,\ell} = T_{t,\ell-1}/\eta$, where $0 < \eta < 1$ (e.g., $\eta = 0.5$).
10:    **while** 1 **do**
11:      $\theta_{t,\ell}^S = \frac{1}{C_\theta} \cdot \theta_{t,\ell-1}^S \odot \exp\left(T_{t,\ell}\left[\Phi^T \frac{x_t}{\Phi\theta_{t,\ell-1}^S} + \frac{\alpha-\mathbb{1}}{\theta_{t,\ell-1}^S}\right]\right)$
12:      **if** $f^S(\theta_{t,\ell}^S) > f^S(\theta_{t,\ell-1}^S) + [\nabla_{\theta_t^S} f^S(\theta_{t,\ell-1}^S)]^T(\theta_{t,\ell}^S - \theta_{t,\ell-1}^S) + \frac{1}{T_{t,\ell}}\Psi(\theta_{t,\ell}^S, \theta_{t,\ell-1}^S)$ **then**
13:        $T_{t,\ell} \leftarrow \eta \cdot T_{t,\ell}$
14:      **else**
15:        break
16:      **end if**
17:    **end while**
18:  **end for**
19:  **for** $a = 1, \ldots, |\mathcal{A}_t|$ **do**
20:    Initialization: $\theta_{t,0}^a = \frac{1}{K}\mathbb{1}$ and $T_{t,0}$.
21:    **for** $\ell = 1, \ldots, L$ **do**
22:      $T_{t,\ell} = T_{t,\ell-1}/\eta$, where $0 < \eta < 1$ (e.g., $\eta = 0.5$).
23:      **while** 1 **do**
24:        $\theta_{t,\ell}^a = \frac{1}{C_\theta} \cdot \theta_{t,\ell-1}^a \odot \exp\left(T_{t,\ell}\left[\Phi^T \frac{x_t}{\Phi\theta_{t,\ell-1}^a} + \frac{\alpha-\mathbb{1}}{\theta_{t,\ell-1}^a}\right]\right)$
25:        **if** $f^a(\theta_{t,\ell}^a) > f^a(\theta_{t,\ell-1}^a) + [\nabla_{\theta_t^a} f^a(\theta_{t,\ell-1}^a)]^T(\theta_{t,\ell}^a - \theta_{t,\ell-1}^a) + \frac{1}{T_{t,\ell}}\Psi(\theta_{t,\ell}^a, \theta_{t,\ell-1}^a)$ **then**
26:          $T_{t,\ell} \leftarrow \eta \cdot T_{t,\ell}$
27:        **else**
28:          break
29:        **end if**
30:      **end while**
31:    **end for**
32:  **end for**
33:  Output: $\hat{\theta}_t^S = \theta_{t,L}^S$, $\hat{\theta}_t^a = \theta_{t,L}^a$, and

$$\mathbb{E}\left\{Q(\theta_t^S, \theta_t^a)|x_{1:t}^S, x_{1:t}^A, a_{1:t-1}\right\} \approx (\theta_{t,L}^a)^T U \theta_{t,L}^S, \quad a = 1, \ldots, |\mathcal{A}_t| \tag{38}$$

34: **end for**

---

- The initial Dirichlet parameters $\alpha_1^S = \alpha_1^A = 1.001$. The rest of the $\alpha_t^S$ and $\alpha_t^A$ is dynamically determined by the model itself and could be less than or greater than one. $\beta_S = \beta_A = 1.001$.

- The discount factor $\gamma = 0.9$, same as the choice in [11].

- $\sigma_r^2 = 3.2$.

- We clip the gradient of $\Phi_S$ and $\Phi_A$ with threshold $10^4$, and we clip the gradients of $W_{SS}, W_{SA}, W_{AS}, W_{AA}$ with threshold 100.

### E.4 Derivation of the Back Propagation Formula

In this appendix, we derive the back propagation formula for learning the LDA model from feedbacks. The cost function of the problem can be expressed as

$$L(\Theta) = \sum_{i=1}^{N} l_i(\Theta) - \ln p(\Theta) = N\left(\frac{1}{N}\sum_{i=1}^{N} l_i(\Theta) - \frac{1}{N}\ln p(\Theta)\right) \tag{39}$$

where

$$l_i(\Theta) \triangleq \sum_{t=1}^{T_i} \frac{1}{2\sigma_r^2} \|d_t - q_t\|^2$$

$$q_t \triangleq (\theta_{t,L}^{a_t})^T U \theta_{t,L}^S$$

$$d_t \triangleq r_t + \gamma \cdot \max_b Q(\theta_{t+1,L}^S, \theta_{t+1,L}^b)$$

The gradients of $\ln p(\Theta)$ with respect to model parameters are relatively easy. Below, we mainly focus on deriving the gradient of $l_i(\Theta)$. We summarize the result before the derivation. Then, we have

$$\Delta q_t = -\frac{1}{\sigma_r^2}(d_t - q_t) \tag{40}$$

$$\frac{\partial l_i}{\partial U} = \sum_{t=1}^{T_i} \Delta q_t \cdot \theta_{t,L}^{a_t}[\theta_{t,L}^S]^T \tag{41}$$

$$\frac{\partial l_i}{\partial \Phi_S} = \sum_{t=1}^{T_i} \sum_{\ell=1}^{L} T_{t,\ell}^S \left\{ \frac{x_t^S}{\Phi_S \theta_{t,\ell-1}^S}(\theta_{t,\ell}^S \odot \xi_{t,\ell}^S)^T - \left[\Phi_S(\theta_{t,\ell}^S \odot \xi_{t,\ell}^S) \odot \frac{x_t^S}{(\Phi_S \theta_{t,\ell-1}^S)^2}\right][\theta_{t,\ell-1}^S]^T \right\} \tag{42}$$

$$\frac{\partial l_i}{\partial \Phi_A} = \sum_{t=1}^{T_i} \sum_{\ell=1}^{L} T_{t,\ell}^{a_t} \left\{ \frac{x_t^{a_t}}{\Phi_A \theta_{t,\ell-1}^{a_t}}(\theta_{t,\ell}^{a_t} \odot \xi_{t,\ell}^{a_t})^T - \left[\Phi_A(\theta_{t,\ell}^{a_t} \odot \xi_{t,\ell}^{a_t}) \odot \frac{x_t^{a_t}}{(\Phi_A \theta_{t,\ell-1}^{a_t})^2}\right][\theta_{t,\ell-1}^{a_t}]^T \right\} \tag{43}$$

$$\frac{\partial l_i}{\partial W_{SS}} = \sum_{t=2}^{T_i} (D_t^S \Delta \alpha_t^S)(\theta_{t-1,L}^S)^T \tag{44}$$

$$\frac{\partial l_i}{\partial W_{SA}} = \sum_{t=2}^{T_i} (D_t^S \Delta \alpha_t^S)(\theta_{t-1,L}^{a_{t-1}})^T \tag{45}$$

$$\frac{\partial l_i}{\partial W_{AS}} = \sum_{t=2}^{T_i} (D_t^A \Delta \alpha_t^A)(\theta_{t-1,L}^S)^T \tag{46}$$

$$\frac{\partial l_i}{\partial W_{AA}} = \sum_{t=2}^{T_i} (D_t^A \Delta \alpha_t^A)(\theta_{t-1,L}^{a_{t-1}})^T \tag{47}$$

$$\xi_{t,\ell-1}^S = (I - \mathbb{1}[\theta_{t,\ell-1}^S]^T) \left\{ \frac{\theta_{t,\ell}^S \odot \xi_{t,\ell}^S}{\theta_{t,\ell-1}^S} \right.$$
$$\left. - T_{t,\ell}^S \left[ \Phi_S^T \mathrm{diag}\left(\frac{x_t^S}{(\Phi_S \theta_{t,\ell-1}^S)^2}\right) \Phi_S + \mathrm{diag}\left(\frac{\alpha_t^S - \mathbb{1}}{(\theta_{t,\ell-1}^S)^2}\right) \right] (\theta_{t,\ell}^S \odot \xi_{t,\ell}^S) \right\} \tag{48}$$

$$\xi_{t,\ell-1}^{a_t} = (I - \mathbb{1}[\theta_{t,\ell-1}^{a_t}]^T) \left\{ \frac{\theta_{t,\ell}^{a_t} \odot \xi_{t,\ell}^{a_t}}{\theta_{t,\ell-1}^{a_t}} \right.$$
$$\left. - T_{t,\ell}^{a_t} \left[ \Phi_A^T \mathrm{diag}\left(\frac{x_t^{a_t}}{(\Phi_A \theta_{t,\ell-1}^{a_t})^2}\right) \Phi_A + \mathrm{diag}\left(\frac{\alpha_t^A - \mathbb{1}}{(\theta_{t,\ell-1}^{a_t})^2}\right) \right] (\theta_{t,\ell}^{a_t} \odot \xi_{t,\ell}^{a_t}) \right\} \tag{49}$$

$$\xi_{t,L}^S = [I - \mathbb{1}(\theta_{t,L}^S)^T]\left(U^T \theta_{t,L}^{a_t} \Delta q_t + W_{SS}^T D_{t+1}^S \Delta \alpha_{t+1}^S + W_{AS}^T D_{t+1}^A \Delta \alpha_{t+1}^A\right) \tag{50}$$

$$\xi_{t,L}^{a_t} = [I - \mathbb{1}(\theta_{t,L}^{a_t})^T]\left(U \theta_{t,L}^S \Delta q_t + W_{SA}^T D_{t+1}^S \Delta \alpha_{t+1}^S + W_{AA}^T D_{t+1}^A \Delta \alpha_{t+1}^A\right) \tag{51}$$

$$\Delta \alpha_t^S = \sum_{\ell=1}^{L} T_{t,\ell}^S \frac{\theta_{t,\ell}^S}{\theta_{t,\ell-1}^S} \odot \xi_{t,\ell}^S, \qquad \Delta \alpha_{T+1}^S = 0 \tag{52}$$

$$\Delta\alpha_t^A = \sum_{\ell=1}^{L} T_{d,\ell}^{a_t} \frac{\theta_{t,\ell}^{a_t}}{\theta_{t,\ell-1}^{a_t}} \odot \xi_{t,\ell}^{a_t}, \qquad \Delta\alpha_{T+1}^A = 0 \tag{53}$$

## E.5   $\Delta q_t$

By the definition of $l_i$ we have

$$\Delta q_t = \frac{\partial l_i}{\partial q_t} = -\frac{1}{\sigma_r^2}(d_t - q_t) \tag{54}$$

## E.6   $\frac{\partial l_i}{\partial U}$

By chain rule, we have

$$\begin{aligned}
\frac{\partial l_i}{\partial U} &= \sum_{t=1}^{T_i} \frac{\partial q_t}{\partial U} \cdot \frac{\partial l_i}{\partial q_t} \\
&= \sum_{t=1}^{T_i} \frac{\partial q_t}{\partial U} \cdot \Delta q_t
\end{aligned} \tag{55}$$

By definition, $q_t = (\theta_{t,L}^{a_t})^T U \theta_{t,L}^S$, we have

$$\frac{\partial q_t}{\partial U} = \theta_{t,L}^{a_t} (\theta_{t,L}^S)^T \tag{56}$$

Substituting the above expression, we arrive at the expression of $\frac{\partial l_i}{\partial U}$.

## E.7   $\frac{\partial l_i}{\partial \Phi_S}$

The expression of $\frac{\partial l_i}{\partial \Phi_S}$ and the recursion of $\xi_{t,\ell}^S$ can be derived in the same manner as that in BP-sLDA. We only derive the expression of $\xi_{t,L}^S$ here. First, note that $\xi_{t,\ell}^S = \mathbb{1}^T p_{t,\ell}^S \cdot \delta_{t,\ell}^S$, where $\delta_{t,\ell}^S \triangleq \frac{\partial l_i}{\partial p_{t,\ell}^S}$. We start by deriving the expression of $\delta_{t,L}^S$. Introduce the notation

$$\Delta\alpha_{t+1}^S = \frac{\partial l_i}{\partial \alpha_{t+1}^S}, \quad \Delta\alpha_{t+1}^A = \frac{\partial l_i}{\partial \alpha_{t+1}^A} \tag{57}$$

Then, we have

$$\begin{aligned}
\delta_{t,L}^S &= \frac{\partial l_i}{\partial p_{t,L}^S} \\
&= \frac{\partial q_t}{\partial p_{t,L}^S} \cdot \frac{\partial l_i}{\partial q_t} + \frac{\partial \alpha_{t+1}^S}{\partial p_{t,L}^S} \cdot \frac{\partial l_i}{\partial \alpha_{t+1}^S} + \frac{\partial \alpha_{t+1}^A}{\partial p_{t,L}^S} \cdot \frac{\partial l_i}{\partial \alpha_{t+1}^A} \\
&= \frac{\partial q_t}{\partial p_{t,L}^S} \cdot \Delta q_t + \frac{\partial \alpha_{t+1}^S}{\partial p_{t,L}^S} \cdot \Delta\alpha_{t+1}^S + \frac{\partial \alpha_{t+1}^A}{\partial p_{t,L}^S} \cdot \Delta\alpha_{t+1}^A
\end{aligned} \tag{58}$$

By the expression of $q_t = (\theta_{t,L}^{a_t})^T U \theta_{t,L}^S$ and $\theta_{t,L}^S = \frac{p_{t,L}^S}{\mathbb{1}^T p_{t,L}^S}$, we have

$$\frac{\partial q_t}{\partial p_{t,L}^S} = \frac{1}{\mathbb{1}^T p_{t,L}^S}(I - \mathbb{1}(\theta_{t,L}^S)^T) U^T \theta_{t,L}^{a_t} \tag{59}$$

Noting that

$$\begin{aligned}
\alpha_{t+1}^S &= \sigma\left(W_{SS}\theta_{t,L}^S + W_{SA}\theta_{t,L}^{a_t} + \alpha_0^S\right) \\
&= \sigma(p_{\alpha_{t+1}^S})
\end{aligned}$$

where

$$p_{\alpha_{t+1}^S} \triangleq W_{SS}\theta_{t,L}^S + W_{SA}\theta_{t,L}^{a_t} + \alpha_0^S$$

we have

$$\frac{\partial(\alpha_{t+1}^S)^T}{\partial p_{t,L}^S} = \frac{\partial(\theta_{t,L}^S)^T}{\partial p_{t,L}^S} \cdot \frac{\partial p_{\alpha_{t+1}^S}^T}{\partial \theta_{t,L}^S} \cdot \frac{\partial(\alpha_t^S)^T)}{\partial p_{\alpha_{t+1}^S}}$$

$$= \frac{1}{\mathbb{1}^T p_{t,L}^S}[I - \mathbb{1}(\theta_{t,L}^S)^T]W_{SS}^T \mathrm{diag}\big(\sigma'(p_{\alpha_{t+1}^S})\big)$$

$$= \frac{1}{\mathbb{1}^T p_{t,L}^S}[I - \mathbb{1}(\theta_{t,L}^S)^T]W_{SS}^T D_{t+1}^S \qquad (60)$$

where

$$D_{t+1}^S \triangleq \mathrm{diag}\big(\sigma'(p_{\alpha_{t+1}^S})\big)$$

Likewise, we can get

$$\frac{\partial(\alpha_{t+1}^A)^T}{\partial p_{t,L}^S} = \frac{1}{\mathbb{1}^T p_{t,L}^S}[I - \mathbb{1}(\theta_{t,L}^S)^T]W_{AS}^T D_{\alpha_{t+1}^A} \qquad (61)$$

where

$$D_{t+1}^A \triangleq \mathrm{diag}\big(\sigma'(p_{\alpha_{t+1}^A})\big)$$
$$p_{\alpha_{t+1}^A} \triangleq W_{AS}\theta_{t,L}^S + W_{AA}\theta_{t,L}^{a_t} + \alpha_0^A \qquad (62)$$

Substituting (59), (60) and (61) into (58), we obtain

$$\delta_{t,L}^S = \frac{1}{\mathbb{1}^T p_{t,L}^S}(I - \mathbb{1}(\theta_{t,L}^S)^T)U^T\theta_{t,L}^{a_t}\Delta q_t$$

$$+ \frac{1}{\mathbb{1}^T p_{t,L}^S}[I - \mathbb{1}(\theta_{t,L}^S)^T]W_{SS}^T D_{t+1}^S \Delta\alpha_{t+1}^S$$

$$+ \frac{1}{\mathbb{1}^T p_{t,L}^S}[I - \mathbb{1}(\theta_{t,L}^S)^T]W_{AS}^T D_{\alpha_{t+1}^A}\Delta\alpha_{t+1}^A$$

Multiplying both sides by $\mathbb{1}^T p_{t,L}^S$, we obtain the desired result.

## E.8 $\frac{\partial l_i}{\partial \Phi_A}$

The related expression for $\frac{\partial l_i}{\partial \Phi_A}$ can be derived in a similar manner as that of $\frac{\partial l_i}{\partial \Phi_S}$. Therefore, we omit the derivation for brevity.

## E.9 $\Delta\alpha_t^S$ and $\Delta\alpha_t^A$

By chain rule, it holds that

$$\Delta\alpha_t^S = \frac{\partial l_i}{\partial \alpha_S}$$

$$= \sum_{\ell=1}^{L} \frac{\partial z_{t,\ell}^S}{\partial \alpha_t^S} \cdot \frac{\partial p_{t,\ell}^S}{\partial z_{t,\ell}^S} \cdot \frac{\partial l_i}{\partial p_{t,\ell}^S}$$

$$= \sum_{\ell=1}^{L} \frac{\partial z_{t,\ell}^S}{\partial \alpha_t^S} \cdot \frac{\partial p_{t,\ell}^S}{\partial z_{t,\ell}^S} \cdot \delta_{t,\ell}^S \qquad (63)$$

Noting that

$$p_{t,\ell}^S = \theta_{t,\ell-1}^S \odot \exp(z_{t,\ell}^S)$$

$$z_{t,\ell}^{S} = T_{t,\ell}^{S} \cdot \left[ (\Phi_{\ell}^{S})^{T} \frac{x_{t}^{S}}{\Phi_{\ell}^{S} \theta_{t,\ell-1}^{S}} + \frac{\alpha_{t}^{S} - \mathbb{1}}{\theta_{t,\ell-1}^{S}} \right] \tag{64}$$

we have

$$\frac{\partial (p_{t,\ell}^{S})^{T}}{\partial z_{t,\ell}^{S}} = \text{diag}\big(\theta_{t,\ell-1}^{S} \odot \exp(z_{t,\ell}^{S})\big) = \text{diag}(p_{t,\ell}^{S})$$

$$\frac{\partial (z_{t,\ell}^{S})^{T}}{\partial \alpha_{t}^{S}} = T_{t,\ell}^{S} \cdot \text{diag} \left( \frac{1}{\theta_{t,\ell-1}^{S}} \right) \tag{65}$$

Substituting the above expressions, we have

$$\Delta \alpha_{t}^{S} = \sum_{\ell=1}^{L} T_{t,\ell}^{S} \cdot \frac{p_{t,\ell}^{S}}{\theta_{t,\ell-1}^{S}} \odot \delta_{t,\ell}^{S}$$

$$= \sum_{\ell=1}^{L} T_{t,\ell}^{S} \cdot \frac{\theta_{t,\ell}^{S}}{\theta_{t,\ell-1}^{S}} \odot \xi_{t,\ell}^{S} \tag{66}$$

where in the last step, we used the fact that $\xi_{t,\ell}^{S} = \mathbb{1}^{T} p_{t,\ell}^{S} \cdot \delta_{t,\ell}^{S}$ and $\theta_{t,\ell}^{S} = \frac{p_{t,\ell}^{S}}{\mathbb{1}^{T} p_{t,\ell}^{S}}$. In a similar manner, we can derive the expression for $\Delta \alpha_{t}^{A}$.

### E.10 $\quad \frac{\partial \ell_i}{\partial W_{SS}}, \frac{\partial l_i}{\partial W_{SA}}, \frac{\partial l_i}{\partial W_{AS}}$ and $\frac{\partial l_i}{\partial W_{AA}}$

We will only derive the expression for $\frac{\partial \ell_i}{\partial W_{SS}}$ and the derivation of the others is similar. Let $[W_{SS}]_{ij}$ denote the $(i,j)$-th component of the matrix $W_{SS}$. Then, by chain rule, we have

$$\frac{\partial l_i}{\partial [W_{SS}]_{ij}} = \frac{\partial \alpha_{t}^{S}}{\partial [W_{SS}]_{ij}} \cdot \frac{\partial l_i}{\partial \alpha_{t}^{S}} = \frac{\partial \alpha_{t}^{S}}{\partial [W_{SS}]_{ij}} \cdot \Delta \alpha_{t}^{S} \tag{67}$$

By the fact that

$$p_{\alpha_{t}^{S}} = W_{SS} \theta_{t-1,L}^{S} + W_{SA} \theta_{t-1,L}^{a_{t-1}} + \alpha_{0}^{S}$$

we have

$$\frac{\partial (\alpha_{t}^{S})^{T}}{\partial [W_{SS}]_{ij}} = \frac{\partial p_{\alpha_{t}^{S}}^{T}}{\partial [W_{SS}]_{ij}} \cdot \frac{\partial (\alpha_{t}^{S})^{T}}{\partial p_{\alpha_{t}^{S}}}$$

$$= \frac{\partial p_{\alpha_{t}^{S}}^{T}}{[W_{SS}]_{ij}} D_{t}^{S}$$

$$= [\theta_{t-1,L}^{S}]_{j} \cdot e_{i}^{T} D_{t}^{S} \tag{68}$$

where $e_i$ is a vector with $i$-th element being one and zero otherwise. Then, it holds that

$$\frac{\partial (\alpha_{t}^{S})^{T}}{\partial [W_{SS}]_{ij}} \cdot \Delta \alpha_{t}^{S} = [\theta_{t-1,L}^{S}]_{j} \cdot [D_{t}^{S}]_{ii} \cdot [\Delta \alpha_{t}^{S}]_{i} \tag{69}$$

so that putting in matrix form:

$$\frac{\partial l_i}{\partial W_{SS}} = (D_{t}^{S} \Delta \alpha_{t}^{S})(\theta_{t-1,L}^{S})^{T}$$