[Reviews · NeurIPS 2017]

Reviewer 1



This paper proposes new reinforcement learning algorithm which can be applied when both action spaces and state spaces are characterised by natural languages. The proposed model combines the LDA and q-learning approaches for this purpose and performs better than the previously suggested models. Overall, this paper is written well and easy to follow. While this paper might be not significantly novel in both topic modelling and reinforcement learning area, the seamless combination of two approaches is still interesting and novel. My biggest concern is the practical use of the proposed model. Although it shows a good performance at the given tasks, making a decision based on topics may not be an ideal idea. For example, what if the possible actions have the same topic distribution? let's say we have two actions: 'go to the coffee shop' and 'go to the restaurant'. The topic of two sentences would be very similar, but the consequence of a different action might be very different. This might be an inherent problem with the suggested method. + missing reference at line 76

Reviewer 2



* Summary This paper introduces the Q-LDA model to learn a sequential decision process for text. The model builds on the LDA topic model to improve the interpretability of the decisions. More specifically, the models draws inspiration from supervised LDA and dynamic LDA. The authors present an algorithm to train the model and apply the model to text-based games. * Evaluation I really liked this paper. It is well written and well explained. I find the problem to be interesting, and the model is interesting as well. I went over the proof of Appendix A and B and I feel like this gave me a good intuition about the proposed training method. The experimental evaluation is a bit weak, but it is not flawed and is interesting, and the model is original enough that it shouldn't be an issue. The proofs are conceptually simple, and the key idea to enable the training is the typical Monte-Carlo integration with 1 sample. * Discussion One thing I missed is why did you choose Q to be of this specific form? It would be great to explain this choice more. I guess we definitely want the theta from the action and the theta from the observation to be involved. What about U? Did you just add such a multiplication to have enough learning potential? Why isn't this going through a non-linearity? * Typos 106: anexploration 113: \alpha_1 or \alpha_t 172: the the a variant Appendix B. You could break down step 29 to 30 a bit more.

Reviewer 3



This paper targets on two text games and propose a new reinforcement learning framework Q-LDA to discover latent patterns in sequential decision process. The proposed model uses LDA to convert action space into a continuous representation and subsequently use Q-learning algorithm to iteratively make decision in a sequential manner. Authors apply the proposed model to two different text games, and achieve better performance than previous proposed baseline models. The paper is a little bit hard to follow with some missing or inconsistent information. The paper is not self-contained, for a reader that is not familiar with the problem domain, one may need to refer to the Appendix or prior works almost all the time. Some detailed comments: - I would suggest authors to include a detailed section highlighting the contribution of the paper. - Authors provide some screenshots on the text game interface in the appendix material, but the information of the text games is still short. The reference [11] also doesn't provide much useful context neither. I would recommend authors to include some example text flow (at least in the Appendix) from these games to better illustrate the target scenario. What are and how many is the possible conversation flow of each text game? - In the context of the game, the agent only receives a reward at the end of game. This is consistent with the text in line 74-75. However, in the graphically model shown in Figure 1. It seems like there is a reward after each turn. I assume this graphical illustration is for general process, but it should be nice to include an explanation in the text. - In the graphically illustration, it is unclear to me which variables are observable and which are not. For example, all \beta_A and \beta_S are not observable. The rewards r_t's are not observable in my understanding. - In the generative process, it occurs to me that the observation text W are generated following LDA process as well, but in the graphically model illustration, there is a missing plate in the figure. - I would encourage authors to include a section of model complexity analysis, for example, what is the complexity of parameter space. Given the size of the dataset and the complexity of the proposed model, it is hard to judge if the learned policy is generalizable. - In the experiments, what are the vocabulary size of the dataset, in terms of observed text and action text? In Table 2 and Figure 2, authors demonstrate some interesting outcome and observation from the generated topics. I am wondering are these topics being used in other episode, since these topics look very fine-grained and hence may not be applicable to other scenario. - I would also recommend to add in a list of all topics (maybe in Appendix), rather than the cherry-picked ones, to display. - In the experiments, authors mention that the upper bound for reward is 20 for "Saving John" and 30 for "Machine of Death". Are these reward numbers objectively specified in the game flow or assigned in a post-hoc manner? If latter, how do you justify this measure (especially if the reward are assigned upon game termination). - Missing reference in line 76. Missing space in line 106. {D}irichlet in line 291. %%%%%%%%%%%%%%%%%%%%%%%% The authors' response have clarified some of my questions and they also agree to improve the paper to make it more self contained. I have adjusted the score accordingly.